# Factors associated with the willingness to provide telerehabilitation by physiotherapists treating older adults or people with neurological diseases during the COVID-19 pandemic in Sweden

Sophia Humphries[1]*, Lucian Bezuidenhout[1,2], Charlotte K. Häger[3], David Moulaee Conradsson[1,4]

**1** Department of Neurobiology, Care Sciences and Society, Karolinska Institute, Stockholm, Sweden, **2** Department of Health and Rehabilitation Sciences, Division of Physiotherapy, Stellenbosch University, Cape Town, South Africa, **3** Department of Community Medicine and Rehabilitation—Physiotherapy Section, Umeå University, Umeå, Sweden, **4** Women's Health and Allied Health Professionals Theme, Medical Unit Occupational Therapy and Physiotherapy, Karolinska University Hospital, Stockholm, Sweden

* sophia.humphries@ki.se

**Data Availability Statement:** Since data can indirectly be traced back to the study participants,

## Abstract

### Background

While telerehabilitation is a promising alternative to traditional rehabilitation, previous studies suggest that it is still underutilised by physiotherapists. The purpose of this study was to identify factors associated with the willingness, and use of, telerehabilitation among physiotherapists.

### Method

An online survey, covering self-reported use of, and attitudes toward telerehabilitation during the COVID-19 pandemic, was distributed to physiotherapists who were members of the Swedish Association of Physiotherapists and working in geriatrics or neurology in Sweden.

### Results

A total of 307 responding physiotherapists were included, most of whom were female (n = 277, 90.2%), working full-time (n = 225, 73.3%), had a bachelor's degree in physiotherapy as their highest education (n = 238, 77.8%) and working in community care settings (n = 131, 43.0%). Overall, 42.3% responded that they would be willing to work with telerehabilitation daily and 47% responded that they had previously worked with telerehabilitation to some degree during the COVID-19 pandemic. Logistic regression analysis revealed that feeling comfortable using digital tools (OR = 1.81, P = .043), believing that telerehabilitation increases the accessibility of rehabilitation (OR = 2.27, P = .009), and that patients will appreciate it (OR = 2.10, P = .025), were significantly associated with willingness to work daily with telerehabilitation. Working in primary care or rehabilitation centres (OR = 3.48, P <

according to the Swedish and EU personal data sharing legislation, access can only be granted upon request from the Research Data Office at Karolinska Institute (rdo@ki.se). Any sharing of data will be regulated via a data transfer and use agreement with the recipient and require ethical approval from the Regional Board of Ethics in Stockholm.

**Funding:** The author(s) received no specific funding for this work.

**Competing interests:** The authors have declared that no competing interests exist.

.012), having previously used telerehabilitation (OR = 55.86, P < .001), and perceiving the workplace reimbursement system as facilitating telerehabilitation (OR = 8.24, P = .003), were factors significantly associated with reported use of telerehabilitation during the COVID-19 pandemic.

## Conclusion

Among physiotherapists in Sweden, willingness to use telerehabilitation is largely associated with personal attitudes towards it, whereas reported use of telerehabilitation appears to be related to organisational factors. These findings could be used to shape future implementation of telerehabilitation practices in geriatric and neurorehabilitation in Sweden and contribute to the broader understanding of telerehabilitation among physiotherapists across different contexts.

### Author summary

Our study explores the acceptance and utilisation of telerehabilitation among physiotherapists in Sweden, focusing on those specialising in geriatrics or neurology. In a survey conducted during the COVID-19 pandemic, 307 physiotherapists, predominantly women (90%), participated. Our findings reveal that the willingness to engage in telerehabilitation is primarily influenced by personal attitudes, such as comfort with digital tools, the belief in increased rehabilitation accessibility, and the perception that patients appreciate this approach. On the other hand, reported usage of telerehabilitation is associated with organisational factors, including working in primary care or rehabilitation centres, prior experience with telerehabilitation, and a supportive workplace reimbursement system. These insights underscore the importance of addressing both individual and organisational aspects to successfully integrate telerehabilitation practices within geriatric and neurorehabilitation settings in Sweden. Our work contributes to the broader understanding of factors shaping the adoption of telerehabilitation, aiming to inform future strategies for its effective implementation.

## Introduction

The COVID-19 pandemic had a significant impact on healthcare, making traditional consultations and assessments impractical or unviable [1]. During the initial wave of the pandemic, the World Health Organization advised the postponement of non-urgent healthcare appointments, prioritizing essential or emergency in-person care [2]. Consequently, many rehabilitation services were temporarily unavailable, despite their critical role in restoring or sustaining the functioning of patients and their participation in society and quality of life [3].

During the COVID-19 pandemic, Swedish authorities opted for a different approach by emphasizing individual responsibility to follow public health recommendations rather than implementing movement restrictions or lockdowns [4]. Specifically, individuals aged ≥ 70 years and people with neurological conditions were identified as high-risk groups and given guidance to follow more stringent social isolation measures, which included reducing social interactions outside their homes and avoiding crowded places and public transport, in contrast to younger adults [5]. Consequently, this resulted in notably limited access to rehabilitation

services for both individuals with neurological conditions and older adults in need [6–8]. This impact was in addition to existing non-pandemic-related barriers to rehabilitation, such as limited transportation options and restricted access to clinics in rural regions [9,10].

As a result of these restrictions on in-person healthcare, there has been a global surge in the use of digital-based healthcare, such as telerehabilitation. Telerehabilitation (TR) involves delivering rehabilitation services remotely using information communication technology (ICT) such as video conferencing and mobile phone applications [11]. In the field of physiotherapy, TR can complement regular rehabilitation services, and previous studies support its effectiveness and feasibility [12–15]. Telerehabilitation has the benefit of reducing the physical burden on patients travelling to clinics and offers flexibility in scheduling for physiotherapists [16]. Research indicates that TR can improve functional outcomes and patient satisfaction in different patient groups, such as people living with neurological diseases or older adults [17,18].

In Sweden, the Swedish Association of Local Authorities and Regions has created a vision (Vision e-Health 2025) to become a global leader in the utilization of e-Health solutions by 2025 [19]. Nevertheless, digital appointment uptake in Sweden has been slow in people above the age of 65 as well as in those with comorbidities [20–22]. As such, not all may stand to benefit from digital healthcare in which access to a stable internet connection and a digital device as well as moderate degree of eHealth literacy is usually required. Access to technology is a major factor affecting access to digital healthcare, with characteristics such as age, ethnicity and education highly associated with both internet use and eHealth literacy among adults [23–25]. Contextual factors such as age-related physical or cognitive impairments also play a large role in digital literacy and digital device use [26]. These factors are important when considering who may be at a disadvantage with the implementation of digital healthcare including TR and why it may be underutilized by patients.

Moreover, TR is underutilized in physiotherapy [27] with some physiotherapists expressing concerns about poor patient compliance and lack of personal interaction with patients [28]. This highlights the need to better understand the perception (e.g., attitudes and willingness) of TR among physiotherapists and to identify strategies to overcome any perceived barriers to adoption. Furthermore, willingness to use TR is a crucial factor when considering the potential barriers to TR implementation; therefore, identifying factors associated with actual use is important in understanding how TR is implemented in healthcare systems. According to a recent study by Albahrough and colleagues, physiotherapists are receptive to the use of TR and believe it can improve accessibility to rehabilitation services [29]. The willingness of physiotherapists to use TR has been associated with factors surrounding its feasibility (such as help desk support and availability of video instructions), easy login, quick solving of IT problems, and documentation [30]. In a recent systematic review of randomized controlled trials (RCTs) and protocols for RCTs investigating factors influencing the delivery of TR, barriers to using TR included those pertaining to ease of use and technology (such as internet connectivity issues) whereas facilitators in its use included the belief that it provided improved access to therapists and that participants appreciated the contact that was provided [31]. To the authors' knowledge, there are no recent studies that have investigated factors related to physiotherapists' willingness to use and reported use of TR in the Swedish context.

This study builds on previous work investigating the use and perceptions of TR by physiotherapists working with people with neurological diseases or older adults during the COVID-19 pandemic in Sweden in 2020 [32]. The results from this study revealed that most of the physiotherapists held positive views towards using ICTs for TR and were willing to work at least weekly with TR. Despite positive views towards using ICTs and willingness to engage in TR, most physiotherapists did not provide TR before or during the COVID-19 pandemic

highlighting a disparity between attitudes and actual usage. Hence, the objective of the current study was to identify factors associated with (1) willingness to use TR and, (2) reported use of TR in clinical practice during the COVID-19 pandemic among physiotherapists in Sweden. Knowledge about factors associated with TR use can inform effective strategies for integrating TR into clinical practice.

## Methods

The data collection in this cross-sectional study was performed using an online survey during the second wave of the COVID-19 pandemic in Sweden (September 2020). The full recruitment and data collection procedure has been described previously [32]. The study was approved by REDACTED and all participants gave informed consent via the online survey before the questions included in the survey were accessed.

### Ethics statement

The study was conducted in accordance with the Declaration of Helsinki and was approved by the Swedish Ethical Review Authority (2020–01.850). All participants provided written informed consent via the online survey before the questions included in the survey were accessed.

### Participants

An email containing the survey link was sent to members of the following special sections of the Swedish Association of Physiotherapists: Neurology, Health of the Elderly, and Primary Care. Approximately 3400 physiotherapists belong to these three target groups out of the total number of approximately 17,000 registered physiotherapists in Sweden [33]. Clinically practicing physiotherapists treating patients with neurological disorders or older adults were included in this study. The full recruitment procedure is described in a previous publication [32].

### Construction of the web survey

The justification for the utilisation of a digital survey was the potential to reach a larger target group in a resource-efficient and rapid manner. The research team adapted this survey from a previous survey regarding physiotherapy for individuals with Parkinson's disease in Sweden [34]. For the aim of this study, additional questions were included to explore the physiotherapists' perceptions of TR, and ICT based on the NASSS (Non-Adopt, Abandon, Scale-Up, Dissemination, Sustainability) framework [35]. NASSS is a framework covering seven domains, including 1) the condition (or illness), 2) the technology, 3) the value proposition, 4) the adopter system, 5) the organisation, 6) the wider context (institutional and societal), and 7) the interaction between these domains over time. NASSS can be used to inform the design of new technology and solutions.

Before disseminating the survey, it was tested among 15 physiotherapists with clinical and/ or research experience in geriatrics and neurorehabilitation for content validity, including usability, readability, and presentation. The survey was then revised based on feedback, which mainly related to terminology; for example, the more familiar term "digital tools" was used instead of ICTs. This process is described in full in a previous publication [32].

### Procedure: Web survey

The web survey was sent via email to the targeted physiotherapists and took approximately ten minutes to complete. The survey was distributed through Typeform (www.typeform.com) and

was open for responders over a five-week period. During this time, three separate reminder emails were distributed.

## Content of the survey

A full breakdown of the items used in the analysis is presented in S1 File and thoroughly described in a previous publication [32].

The first part of the survey covered demographic and workplace factors. The remaining questions focused on 1) the use of TR before and during the COVID-19 pandemic (i.e., the proportion of patients receiving TR), 2) willingness to use TR (i.e., the amount of time the physiotherapists would be willing to devote to TR) and 3) the physiotherapist's perception of TR. The latter domain was further divided into questions covering the physiotherapist's own perspective (e.g., comfort; "I am comfortable using digital tools for telerehabilitation"), how the physiotherapists perceive their patients' experience of TR services (e.g., appreciation; "I think patients will appreciate telerehabilitation") and the workplace (e.g., workplace reimbursement for telerehabilitation; "The reimbursement system that applies to my workplace facilitates telerehabilitation with the support of digital tools").

## Data analysis

Data analysis was performed using R version 4.2.1. [36]. All data were categorical and either binomial or ordinal. Demographic and workplace factors were presented descriptively as numbers and percentages. The reported modes of TR that were utilized, including telephone communication (verbal), short message communication (SMS) services, video conferencing, internet-based applications, and mobile applications, were presented descriptively.

Factors associated with willingness to use TR and use of TR during the COVID-19 pandemic were analysed using two multiple logistic regression models. In the first model, the dependent variable willingness to use TR was analysed using the question "*How much of your workday would you consider devoting to TR using digital tools*?" to which participants responded using a 5-point Likert scale (i.e., 1) "entire working day", 2) "half working day", 3) "1–2 hours per week", 4) "a few times per week" and 5) "not at all". Responses 1–3 were classified as being willing to use TR in clinical practice. In the second model, the variable use of TR was obtained through responses to the question "*Which statement about telerehabilitation best reflects your work with patients during the COVID-19 pandemic*?" using a 4-point Likert scale (i.e., 1) "all rehabilitation is performed remotely", 2) "about half of patients are treated remotely", 3) "a few patients are treated remotely" and 4) "no patients are treated remotely". Responses 1–3 were classified as using TR during the COVID-19 pandemic.

Covariates included in both regression models were sex, age (20–39 years, 40–49 years, above 50), highest education achieved (bachelor's degree, graduate level or higher), work experience (less than 10 years, 10–19 years, more than 19 years), work setting (community care, hospital, primary care, and rehabilitation centres) and previous use of TR (yes/no). Covariates pertaining to the perception of TR included in the models were dichotomized into two levels (agree/disagree) and presented in S1 File.

Covariates were first modelled as unadjusted univariate logistic regression models. An alpha level $P \leq 0.1$ was considered the cut-off to include factors in the multivariate analyses. Prior to multivariable analyses, measures of association, i.e., chi-Square tests, were computed to assess collinearity among potential associated factors. For the univariate and multivariate regression analyses, effect estimates are presented as adjusted odds ratios (ORs) for the univariate and multiple logistic models and with 95% confidence interval values. $P < 0.05$ was considered statistically significant for all tests.

## Results

A total of 385 physiotherapists responded to the survey, of which 307 were included in the study. Those not working with people with neurological diseases or older adults (n = 68) or working clinically (n = 10) were excluded. Demographic and workplace factors are presented in Table 1. Most respondents were female (n = 277, 90.2%), working full-time (n = 225, 73.3%), had a bachelor's degree in physiotherapy as their highest education (n = 238, 77.8%) and had reported >19 years of clinical work experience (n = 157, 51.1%). Most respondents were working in community care settings (n = 131, 43.0%) with >5 physiotherapists employed (n = 175, 57.0%).

### Factors associated with willingness to use telerehabilitation in clinical practice

As shown in Fig 1, approximately four out of ten responding physiotherapists (42.3%) were willing to work daily (i.e., at least 1–2 hours per day) with TR.

Univariate logistic regression analyses (Table 2) revealed that male sex (OR = 1.90, $P <$ .100), younger age (OR = 2.63, $P <$ .001), less than 10 years of clinical work experience (OR = 2.25, $P <$ .005), working in primary care or rehabilitation centres (OR = 1.87, $P$ = .031) and reporting previous use of TR (OR = 1.72, $P$ = .041) were factors positively associated with willingness to work daily with TR. In addition, all variables relating to the perception of TR, except patient access, were positively associated with willingness to work daily with TR (Table 2). The multivariable logistic regression model (Table 2) showed that physiotherapists

**Table 1. Background and workplace demographics of the survey respondents.**

|  | N = 307 |  |
| --- | :---: | :---: |
| **Background and workplace demographics** | **N (%)** | **Missing (n)** |
| *Female* | 277 (90.2) |  |
| *Age (years)* |  |  |
| 20–39 | 89 (29.0) |  |
| 40–49 | 79 (25.7) |  |
| >50 | 139 (45.3) |  |
| *Working full-time* | 225 (73.3) |  |
| *Highest education achieved* |  | 1 |
| Bachelor's degree | 238 (77.8) |  |
| Graduate level or higher | 68 (22.2) |  |
| *Work experience (years)* |  |  |
| Less than 10 | 72 (23.5) |  |
| 10–19 | 78 (25.4) |  |
| More than 19 | 157 (51.1) |  |
| *Primary patient group* |  |  |
| Geriatric | 168 (54.7) |  |
| Neurology | 139 (45.3) |  |
| *Work setting* |  | 2 |
| Community care | 131 (43.0) |  |
| Hospital | 96 (31.5) |  |
| Primary care and rehabilitation centres | 78 (25.6) |  |
| *Physiotherapists in the workplace* |  |  |
| I work alone | 20 (6.5) |  |
| 1–5 | 112 (36.5) |  |
| >5 | 175 (57.0) |  |

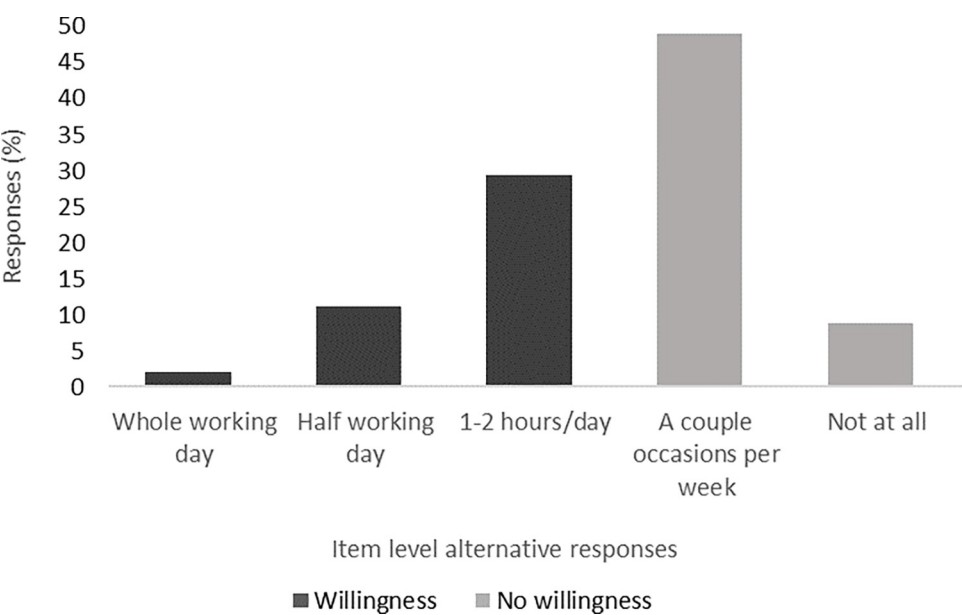

**Fig 1. Willingness to use telerehabilitation among physiotherapists working with individuals with neurological disease or older adults.**

who felt comfortable using technology (OR = 1.81, $P$ = .043), perceived that technology would increase accessibility (OR = 2.27, $P$ = .009) and that patients would appreciate TR (OR = 2.10, $P$ = .025) remained independently associated with willingness to work with TR (Nagelkerke's pseudo $R^2$: .30).

## Factors associated with use of telerehabilitation during the COVID-19 pandemic

As shown in Fig 2, almost half of the responding physiotherapists (47%) reported that they provided TR for their patients to some degree during the COVID-19 pandemic. Most of the frequent ICT use (at least weekly) was provided in the form of telephone communication (78%) (i.e., appointment bookings, provision of follow-ups and advice giving), followed by internet-based applications (14%). Video conferencing was used by 32% of respondents several days per month, making it the most used ICT.

Univariate logistic regression analyses (Table 2) revealed that physiotherapists who had a graduate level education or higher (OR = 2.17, $P$ < .006), worked in a hospital (OR = 1.71, $P$ = .054) or primary care or rehabilitation centres (OR = 6.68, $P$ < .001) and reported previous use of TR before the pandemic (OR = 54.27, $P$ < .001) were more likely to report use of TR ($P$ < .10). All variables measuring the perception of TR (except comfort) were positively associated with the use of TR in clinical practice (Table 2). The multivariable logistic regression model (Table 2) showed that working in primary care and rehabilitation centres (OR = 3.48, $P$ = .012), having previously used TR (OR = 55.86, $P$ < .001) and perceiving the workplace reimbursement system as facilitating TR (OR = 8.24, $P$ = .003) remained independently associated with the use of TR (Nagelkerke's pseudo $R^2$ .58).

## Discussion

This current study investigated factors (i.e., personal and workplace-related) associated with the willingness to use TR and the use of TR during the COVID-19 pandemic among

**Table 2. Univariate and multivariate logistic regression models for willingness to use telerehabilitation provision of telerehabilitation during the COVID-19 pandemic.**

| Variable | Willingness to use telerehabilitation | | | | Provision of telerehabilitation during the COVID-19 pandemic | | | |
|---|---|---|---|---|---|---|---|---|
| | Univariate models | | Multivariate model | | Univariate models | | Multivariate model | |
| | Adjusted OR (95% CI) | P value | Adjusted OR (95% CI) | P value | Adjusted OR (95% CI) | P value | Adjusted OR (95% CI) | P value |
| *Sex* (ref. female) | | | | | | | | |
| Male | 1.90 (0.89–4.13) | .099 | 2.34 (0.95–5.79) | .065 | 0.71 (0.32–1.51) | .376 | | |
| *Age* (ref. >50) | | | | | | | | |
| 20–39 | 2.63 (1.53–4.57) | < .001 | 1.65 (0.60–4.53) | .332 | 0.86 (0.50–1.47) | .581 | | |
| 40–49 | 1.33 (0.75–2.36) | .323 | 1.16 (0.57–2.38) | .677 | 1.23 (0.70–2.15) | .471 | | |
| *Highest education achieved* (ref. bachelor's degree) | | | | | | | | |
| Graduate level or higher | 0.69 (0.39–1.19) | .195 | | | 2.17 (1.25–3.84) | .006 | 1.63 (0.75–3.55) | .221 |
| *Work experience (ref >19 years)* | | | | | | | | |
| Less than 10 years | 2.25 (1.28–4.00) | .005 | 1.34 (0.47–3.76) | .584 | 0.74 (0.42–1.30) | .294 | | |
| 10–19 years | 1.39 (0.80–2.43) | .240 | 1.12 (0.50–2.47) | .787 | 0.67 (0.38–1.17) | .161 | | |
| *Primary patient group* (ref. geriatric) | | | | | | | | |
| Neurology | 0.86 (0.54–1.35) | .507 | | | 1.15 (0.73–1.82) | .536 | | |
| *Work setting* (ref. community care) | | | | | | | | |
| Hospital | 0.83 (0.48–1.43) | .512 | 0.84 (0.44–1.61) | .598 | 1.71 (0.99–2.97) | .054 | 1.49 (0.64–3.48) | .356 |
| Primary care and rehabilitation centres | 1.87 (1.06–3.31) | .031 | 1.02 (0.47–2.22) | .957 | 6.68 (3.56–12.99) | < .001 | 3.48 (1.3–9.22) | .012* |
| *Previous use* (ref. no) | | | | | | | | |
| Yes | 1.72 (1.02–2.90) | .041 | 1.17 (0.62–2.24) | .627 | 54.27 (19.37–226.79) | < .001 | 55.86 (16.04–194.58) | < .001* |
| *Personal views* (ref. no) | | | | | | | | |
| Comfort[a] | 2.86 (1.77–4.68) | < .001 | 1.81 (1.02–3.22) | .043* | 1.17 (0.74–1.85) | .512 | | |
| Accessibility[b] | 4.22 (2.59–6.97) | < .001 | 2.27 (1.23–4.20) | .009* | 1.52 (0.97 2.40) | .072 | 0.89 (0.41–1.94) | .767 |
| Patient appreciation[c] | 4.30(2.66–7.05) | < .001 | 2.10 (1.10–4.01) | .025* | 2.17 (1.36–3.49) | .001 | 1.14 (0.49–2.66) | .756 |
| Patient capability[d] | 3.29 (1.79–6.26) | < .001 | 1.68 (0.79–3.59) | .177 | 2.83 (1.51–5.50) | .002 | 1.79 (0.67–4.77) | .242 |
| Patient access[e] | 1.42 (0.90–2.25) | .130 | | | 2.37 (1.50–3.79) | < .001 | 1.28 (0.60–2.73) | .523 |
| Workplace access[f] | 1.86 (1.18–2.96) | .008 | 1.27 (0.71–2.26) | .421 | 2.17 (1.37–3.46) | .001 | 1.53 (0.77–3.04) | .225 |
| Supportive colleagues[g] | 1.80 (1.14–2.88) | .013 | 1.17 (0.65–2.10) | .606 | 2.65 (1.66–4.26) | < .001 | 1.19 (0.59–2.37) | .629 |
| Financial reimbursement | 2.71 (1.43–5.28) | .003 | 1.98 (0.82–4.77) | .129 | 21.71 (7.65–91.26) | < .001 | 8.24 (2.08–32.59) | .003* |

*p < .05 on the multivariate level

[a] I am comfortable using digital tools for digital rehabilitation

[b] I believe that digital tools will increase the accessibility of rehabilitation

[c] I think patients will appreciate telerehabilitation

[d] Most patients I treat can use digital tools (e.g. computer, tablet, or mobile application) in their rehabilitation

[e] Most patients I treat have access to a computer, tablet, or mobile phone

[f] My workplace has access to digital tools for telerehabilitation

[g] I perceive support from my colleagues and my boss regarding new digital tools for telerehabilitation

[h] The reimbursement system that applies to my workplace facilitates telerehabilitation with the support of digital tools

physiotherapists in Sweden. Physiotherapists' perceptions of TR (i.e., comfort using technology, perceived increased accessibility and patient appreciation) were the strongest factors associated with willingness to use TR, whereas reported use of TR was largely associated with organisational factors (i.e., work setting, previous use of TR and perception of the workplace

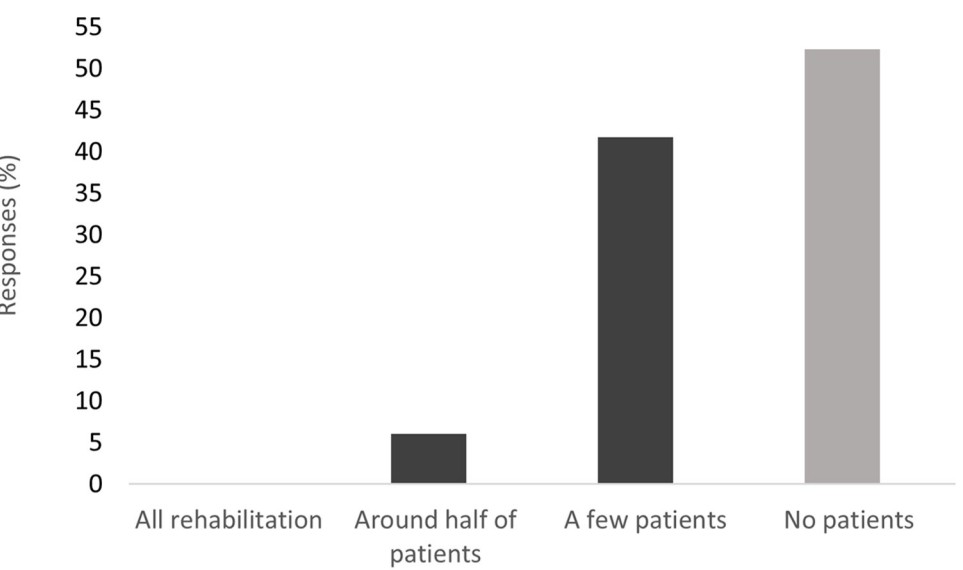

**Fig 2. Use of telerehabilitation during the COVID-19 pandemic among physiotherapists working with individuals with neurological disease or older adults.**

reimbursement system as facilitating TR). These findings could inform future strategies aimed at improving the implementation of TR in the Swedish healthcare setting. The findings of this study also broaden our understanding of the factors influencing the adoption of telerehabilitation, offering insights that could guide its implementation in diverse contextual settings.

The present results of the association between the physiotherapists' positive perception of TR and willingness to use TR are similar to previous findings on physiotherapists in Kuwait, Greece, and India [28,29,37]. Interestingly, two of the three factors associated with TR willingness in our study were patient-related, i.e., how physiotherapists perceive patient reception of TR. Few studies have explored the health professional perspective of patient receival when investigating willingness to use TR. Brouns et al. [30] did report perceived patient benefits among healthcare professionals as a factor that positively influenced willingness to use TR, which could explain why these factors were linked to TR willingness in the present study. Interestingly, willingness to use TR was not associated with age or education level, unlike similar studies' findings [29,38]. The fact that the age distribution of our sample was heavily skewed towards those aged >50 could partly explain this difference but not the association solely with variables measuring perception of TR. However, these findings are supportive of similar studies conducted in Kuwait and Greece in which facilitating factors for the use of TR, from the perspective of physiotherapists, were found to include motivation and interest in TR, as well as patient satisfaction [29,37]. Consistent with this finding is also patients' own willingness to use and attitudes towards TR as an alternative to rehabilitation services in stroke survivors [38]. Supported by our results, previous studies also found comfort with using ICTs for TR to be important for both physiotherapists and patients [28,38].

In the online survey we did not include a question about the physiotherapist's concerns, or their perception of their patients' concerns, towards safety related to TR compared to traditional rehabilitation. Concerns for safety is sometimes reported as a barrier towards

telerehabilitation from patients [39], while on the other hand feasibility studies of telerehabilitation following stroke have reported good feasibility in relation to safety [40,41]. Without having investigated attitudes towards the safety of telerehabilitation specifically it remains unknown whether this could have been a factor contributing towards the willingness to use digital tools.

In total, just under half of the respondents reported using TR during the COVID-19 pandemic which is consistent with studies that reported the use of TR during the COVID-19 pandemic, across several health professions and countries [37,42]. Telephone communication was the most frequently used ICT for those reporting having used TR during the COVID-19 pandemic, most likely due to its ease of use and minimal technical requirements, followed by internet applications. Our finding that physiotherapists who had previously used TR were more likely to report having used it during the pandemic is also in line with previous studies investigating factors related to using digital health tools in patient care [43,44].

Physiotherapists working in primary care and perceiving their workplace reimbursement system as facilitating TR were more likely to have used TR during the COVID-19 pandemic, suggesting that the workplace setting plays a large role in TR use among the physiotherapists in our sample. In line with this finding, barriers to using ICTs in healthcare have covered workplace-related aspects, including the lack of an established compensation system for digital treatment [45]. Consistent with this, a large proportion of healthcare professionals across a variety of work settings in Germany reported regulatory or technical obstacles to the implementation of ICT and remote health care services [46]. This would suggest that organisations present the greatest opportunity to overcome the obstacles that lie in the way of the implementation of digital health treatments such as TR.

## Methodological considerations

A limitation of our study is the potential for selection bias due to the relatively small number of physiotherapists who responded to the survey. The sample in our study, while small, was an approximate representation of females in the target population of physiotherapists [34]. On the other hand, the sample in our study seems to be over representative of the age category >50 years which was the largest age group in our sample and, according to the same study, only accounts for approximately 30% of the total target population [34].

Furthermore, an important factor that should be brought to attention within our findings is the difference between physiotherapists' perceptions of their patients and the actual views and beliefs of their patients. Since the survey was focused on only physiotherapists' attitudes and use of TR, it is not possible to know whether the opinions of the physiotherapists' patient perspectives reflected the actual perspective of the patients they treat or how well they align with the beliefs of their patients. The belief that TR can increase accessibility for patients and that patients will appreciate TR should therefore be emphasized from a physiotherapist and not a patient perspective, although there is some evidence to suggest that patients hold similar attitudes toward the benefits of TR [38]. In further support of this, user satisfaction in one study was very similar for both patients and therapists in multiple aspects of the TR (e.g., ease of use, suitability, and acceptance) [47], so while we cannot be sure that patients would hold equally positive views towards it as the physiotherapists, we can assume partial substantiation.

In our study, willingness was dichotomized into two levels that reflected the willingness to work with TR daily (willing to devote at least 1–2 hours per day to TR) or not (willing to work with TR on a couple of occasions per week, or not at all). It should be noted that the amount of time one is willing to spend using TR might not be the only measure of overall willingness to use TR in clinical practice. Willingness in the study by Albahrouh and Buabbas [29] was a

domain that covered several aspects that the present study did not consider (such as willingness to use TR to obtain consultations from other hospitals or use TR to watch live physiotherapy sessions), making willingness to use TR a broader concept in their findings and potentially a limiting one in the present study.

While our results were obtained in a Swedish setting our findings mirror those of similar studies conducted in other countries, highlighting the importance of our findings across different contexts [29,37]. Our findings suggest the factors associated with the use of TR are largely contextual and related to the work setting. If implementing TR in clinical practice, it could be of benefit for work providers to properly communicate the policy on reimbursement of TR, since this was a perception of a workplace factor that was heavily related to use of TR. Our concluding advice would be that the integration of TR into organisational structures needs to be properly investigated in the appropriate context. Delivery of TR is arguably too complex to narrow down to a single domain, such as patients or providers, but instead should focus on several domains independently, as well as their intersection with each other [11]. Our findings suggest that while physiotherapists, in general, are willing to provide TR, policy and work care settings are what currently dictate the actual provision of TR.

## Conclusion

The present results showed that willingness to use TR among physiotherapists in neurology and geriatrics was largely associated with comfort using ICTs for TR and the belief that they increase rehabilitation accessibility, as well as positively perceiving that their patients will appreciate TR. On the other hand, the reported use of TR was largely associated with organisational factors. For factors related to willingness to use TR, such as comfort using ICTs, organisations could support physiotherapists by providing educational training in these areas as well as addressing the reimbursement conditions that apply and any knowledge barriers surrounding this.

## Supporting information

**S1 File. Variables and criteria used for categorization of the independent and dependent variables.**
(DOCX)

## Acknowledgments

We would like to thank the respondents of this survey for their time and input.

## Author Contributions

**Conceptualization:** Sophia Humphries, Lucian Bezuidenhout, David Moulaee Conradsson.

**Formal analysis:** Sophia Humphries.

**Supervision:** David Moulaee Conradsson.

**Visualization:** Sophia Humphries, Lucian Bezuidenhout, Charlotte K. Häger, David Moulaee Conradsson.

**Writing – original draft:** Sophia Humphries.

**Writing – review & editing:** Sophia Humphries, Lucian Bezuidenhout, Charlotte K. Häger, David Moulaee Conradsson.

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
