## [Decision Letter · Decision Letter 0]

9 Jun 2024

PDIG-D-24-00147

Factors associated with the willingness to provide telerehabilitation by physiotherapists treating older adults or people with neurological diseases during the COVID-19 pandemic in Sweden

PLOS Digital Health

Dear Dr. Humphries,

Thank you for submitting your manuscript to PLOS Digital Health. After careful consideration, we feel that it has merit but does not fully meet PLOS Digital Health's publication criteria as it currently stands. Therefore, we invite you to submit a revised version of the manuscript that addresses the points raised during the review process.

Please submit your revised manuscript within 60 days Aug 08 2024 11:59PM. If you will need more time than this to complete your revisions, please reply to this message or contact the journal office at digitalhealth@plos.org. Please include the following items when submitting your revised manuscript:

We look forward to receiving your revised manuscript.

Kind regards,

Calvin Or, PhD

Section Editor

PLOS Digital Health

Journal Requirements:

1. Please send a completed 'Competing Interests' statement, including any COIs declared by your co-authors. If you have no competing interests to declare, please state "The authors have declared that no competing interests exist". Otherwise please declare all competing interests beginning with the statement "I have read the journal's policy and the authors of this manuscript have the following competing interests:"

2. Please provide separate figure files in .tif or .eps format only and remove any figures embedded in your manuscript file. Please also ensure that all files are under our size limit of 10MB.

Additional Editor Comments (if provided):

Reviewers' comments:

Reviewer's Responses to Questions

**Comments to the Author**

1. Does this manuscript meet PLOS Digital Health’s publication criteria? Is the manuscript technically sound, and do the data support the conclusions? The manuscript must describe methodologically and ethically rigorous research with conclusions that are appropriately drawn based on the data presented.

Reviewer #1: Yes

Reviewer #2: Yes

Reviewer #3: Yes

2. Has the statistical analysis been performed appropriately and rigorously?

Reviewer #1: Yes

Reviewer #2: Yes

Reviewer #3: Yes

3. Have the authors made all data underlying the findings in their manuscript fully available (please refer to the Data Availability Statement at the start of the manuscript PDF file)?

Reviewer #1: Yes

Reviewer #2: Yes

Reviewer #3: No

4. Is the manuscript presented in an intelligible fashion and written in standard English?

Reviewer #1: Yes

Reviewer #2: Yes

Reviewer #3: Yes

5. Review Comments to the Author

Reviewer #1: I have no major issues with this manuscript.

The study aimed to identify factors associated with (i) willingness to use tele-rehabilitation and, (ii) reported use of tele-rehabilitation in clinical practice during the COVID-19 pandemic among physiotherapists (PTs) in Sweden and achieves these objectives. Although pertaining to only Sweden, the results of the study could be useful to the wider physiotherapy profession globally, in informing future work in this area. The use of digital tools is becoming more popular and as such, provides some insights into this for PTs. 

The introduction is informative, and acknowledges the benefits to patients, but nothing about risks/drawbacks. I would like to see an mentioned aspects that do not benefit patients and can be exclusive, for example digital poverty. Not everyone has access and can do this type of rehabilitation. This is alluded to in the results when they refer to "patient access"(line 246), but should be acknowledged.

The methods are generally well written. It is helpful to refer to the previous publication of the recruitment and study data collection methods which have been previously published. Figures and tables are clear and appropriate. Results are clear.

In the discussion, it is interesting that the willingness to use digital tools, did not include concerns for patient safety from PTs, but patient appreciation. Was patient safety mentioned at all in the questionnaire; could this be discussed as in previous literature. 

Conclusion is appropriate.

Limitations are appropriate.

Reviewer #2: Clinically relevant paper finding that among physiotherapists in Sweden, willingness to use tele-rehabilitation is largely associated with personal attitudes towards it, whereas reported use of tele-rehabilitation appears to be related to 

organisational factors. These findings could be used to shape future implementation of tele-rehabilitation practices in geriatric and neurorehabilitation in Sweden and contribute to the broader understanding of tele-rehabilitation 

among physiotherapists across different contexts.

Reviewer #3: A well written manuscript with novel and interesting findings presented. Statistical analyses are performed appropriately and rigourously with figures and tables presented to support claims. Conclusions are supported by the data. In response to Q3, authors have specified that data is not able to be publicly shared due to participant privacy, but data will be made available upon request.

6. PLOS authors have the option to publish the peer review history of their article (what does this mean?). If published, this will include your full peer review and any attached files.

**Do you want your identity to be public for this peer review?** For information about this choice, including consent withdrawal, please see our Privacy Policy.

Reviewer #1: No

Reviewer #2: No

Reviewer #3: Yes: Nicki Newton

---

## [Decision Letter · Decision Letter 1]

27 Jun 2024

Factors associated with the willingness to provide telerehabilitation by physiotherapists treating older adults or people with neurological diseases during the COVID-19 pandemic in Sweden

PDIG-D-24-00147R1

Dear Miss Humphries,

We are pleased to inform you that your manuscript 'Factors associated with the willingness to provide telerehabilitation by physiotherapists treating older adults or people with neurological diseases during the COVID-19 pandemic in Sweden' has been provisionally accepted for publication in PLOS Digital Health.

Best regards,

Calvin Or, PhD

Section Editor

PLOS Digital Health

Reviewer Comments (if any, and for reference):

Reviewer's Responses to Questions

**Comments to the Author**

1. If the authors have adequately addressed your comments raised in a previous round of review and you feel that this manuscript is now acceptable for publication, you may indicate that here to bypass the “Comments to the Author” section, enter your conflict of interest statement in the “Confidential to Editor” section, and submit your "Accept" recommendation.

Reviewer #1: All comments have been addressed

Reviewer #2: All comments have been addressed

Reviewer #3: All comments have been addressed

2. Does this manuscript meet PLOS Digital Health’s publication criteria? Is the manuscript technically sound, and do the data support the conclusions? The manuscript must describe methodologically and ethically rigorous research with conclusions that are appropriately drawn based on the data presented.

Reviewer #1: Yes

Reviewer #2: Yes

Reviewer #3: Yes

3. Has the statistical analysis been performed appropriately and rigorously?

Reviewer #1: Yes

Reviewer #2: N/A

Reviewer #3: Yes

4. Have the authors made all data underlying the findings in their manuscript fully available (please refer to the Data Availability Statement at the start of the manuscript PDF file)?

Reviewer #1: Yes

Reviewer #2: Yes

Reviewer #3: Yes

5. Is the manuscript presented in an intelligible fashion and written in standard English?

Reviewer #1: Yes

Reviewer #2: Yes

Reviewer #3: Yes

6. Review Comments to the Author

Reviewer #1: Comments addressed satisfactorily.

Reviewer #2: I am happy with the revisions made to the manuscript. This is an interesting and clinically relevant paper.

I am happy with the revisions made to the manuscript. This is an interesting and clinically relevant paper.

Reviewer #3: The authors have adequately addressed all comments.

7. PLOS authors have the option to publish the peer review history of their article (what does this mean?). If published, this will include your full peer review and any attached files.

**Do you want your identity to be public for this peer review?** For information about this choice, including consent withdrawal, please see our Privacy Policy.

Reviewer #1: No

Reviewer #2: **Yes: **

Reviewer #3: No
